# Estimated impact of the COVID-19 pandemic on cancer services and excess 1-year mortality in people with cancer and multimorbidity: near real-time data on cancer care, cancer deaths and a population-based cohort study

Alvina G Lai ,[1,2] Laura Pasea,[1,2] Amitava Banerjee ,[1,2,3] Geoff Hall,[4,5,6] Spiros Denaxas,[1,2,7,8] Wai Hoong Chang,[1,2] Michail Katsoulis,[1] Bryan Williams ,[7,9,10] Deenan Pillay,[11] Mahdad Noursadeghi,[11] David Linch,[7,12] Derralynn Hughes,[13,14] Martin D Forster,[10,13] Clare Turnbull ,[15] Natalie K Fitzpatrick,[1,2] Kathryn Boyd,[16] Graham R Foster,[17] Tariq Enver,[13] Vahe Nafilyan ,[18] Ben Humberstone,[18] Richard D Neal,[19] Matt Cooper,[4,5] Monica Jones,[4,5] Kathy Pritchard-Jones,[4,20,21,22] Richard Sullivan,[23] Charlie Davie,[4,14,20] Mark Lawler,[4,24] Harry Hemingway [1,2,7]

AL, ML and HH are joint senior authors.

For numbered affiliations see end of article.

**Correspondence to**
Dr Alvina G Lai;
alvina.lai@ucl.ac.uk

## ABSTRACT

**Objectives** To estimate the impact of the COVID-19 pandemic on cancer care services and overall (direct and indirect) excess deaths in people with cancer.

**Methods** We employed near real-time weekly data on cancer care to determine the adverse effect of the pandemic on cancer services. We also used these data, together with national death registrations until June 2020 to model deaths, in excess of background (pre-COVID-19) mortality, in people with cancer. Background mortality risks for 24 cancers with and without COVID-19-relevant comorbidities were obtained from population-based primary care cohort (Clinical Practice Research Datalink) on 3 862 012 adults in England.

**Results** Declines in urgent referrals (median=−70.4%) and chemotherapy attendances (median=−41.5%) to a nadir (lowest point) in the pandemic were observed. By 31 May, these declines have only partially recovered; urgent referrals (median=−44.5%) and chemotherapy attendances (median=−31.2%). There were short-term excess death registrations for cancer (without COVID-19), with peak relative risk (RR) of 1.17 at week ending on 3 April. The peak RR for all-cause deaths was 2.1 from week ending on 17 April. Based on these findings and recent literature, we modelled 40% and 80% of cancer patients being affected by the pandemic in the long-term. At 40% affected, we estimated 1-year total (direct and indirect) excess deaths in people with cancer as between 7165 and 17 910, using RRs of 1.2 and 1.5, respectively, where 78% of excess deaths occured in patients with ≥1 comorbidity.

**Conclusions** Dramatic reductions were detected in the demand for, and supply of, cancer services which have not fully recovered with lockdown easing. These may contribute, over a 1-year time horizon, to substantial

## Strengths and limitations of this study

► This is the first study that used hospital data and a predictive model to dissect and quantify the adverse impact on mortality of the pandemic on patients with cancer and multimorbidity.

► This study used the breadth of longitudinal information in primary care records from the Clinical Practice Research Datalink to generate background (pre-COVID-19) mortality estimates for patients with cancer.

► This study generated 1-year mortality estimates for 24 cancer types and evaluated the extent by which multimorbidity influences mortality risk in patients with cancer. We considered 15 comorbidity clusters, which include 40 non-malignant comorbidities defined by the Public Health England as associated with severe and fatal COVID-19 infection.

► This study modelled excess deaths using information on background mortality risk and plausible relative risk estimates obtained from the Office for National Statistics and other published studies.

► A limitation of this study is the use of primary care health records which may have missed cases of cancer resulting in more conservative estimations of excess deaths.

excess mortality among people with cancer and multimorbidity. It is urgent to understand how the recovery of general practitioner, oncology and other hospital services might best mitigate these long-term excess mortality risks.

## INTRODUCTION

The COVID-19 pandemic may cause additional (excess) deaths due to both the direct effects of infection and the indirect effects that result from the repurposing of health services designed to address the pandemic.[1] People with cancer are at increased risk of contracting and dying from SARS-CoV-2 infection.[2 3] Optimal cancer care must balance protecting patients from SARS-CoV-2 infection with the need for continued access to early diagnosis and delivery of optimal treatment.[4 5] Professional cancer associations internationally have recommended reducing systemic anticancer treatment, surgery and risk-adapted radiotherapy.[6] In June 2020, the National Health Service (NHS) released statistics for April 2020, indicating that referrals to a consultant for urgent diagnosis of cancer had fallen by 60%.[7] Some cancer surgical procedures have been postponed and cancer screening programmes paused.[8–13]

However, COVID-19-induced healthcare service reconfiguration and recovery have, to date, not been informed by the near real-time hospital data quantifying the extent of disruption for patients with cancer resulting from this service reconfiguration, nor its impact on excess deaths in people with cancer. A previous study has employed literature-based estimates to model the impact of potential diagnostic delays in colorectal cancer during the COVID-19 pandemic.[14 15] Short-term (30 days) death in people with cancer and COVID-19 is importantly driven by (treatable) comorbidities such as hypertension and cardiovascular disease.[16] Public Health England (PHE) has identified patients with these and a wide range of other non-malignant conditions at greater risk of developing severe illness from SARS-CoV-2 exposure,[17 18] while multimorbidity in cancer is an increasing clinical concern.[19 20] For general practitioners and oncologists, evidence is required on the pan-cancer estimation of mortality risks according to type and number of comorbid conditions. Such evidence may inform individual decisions about physical isolation and shielding, as well as the need to ensure that patients access specialist cancer care and seek preventive care for non-malignant comorbidities.

Our objectives were: (1) to quantify changes in cancer care, reporting near real-time weekly data (to June 2020) for urgent referral (for early diagnosis of cancer) and chemotherapy attendance (for treatment of cancer); (2) to quantify short-term direct and indirect excess deaths using near real-time weekly death registrations from the Office for National Statistics (ONS); (3) to estimate the number of annual direct (COVID-19) and indirect excess deaths using population-based 1-year Kaplan-Meier mortality estimates for 24 cancer types and (4) to determine the extent by which multimorbidity contributes to these excess deaths.

## METHODS
### Weekly near real-time hospital data
To estimate the extent to which changes in cancer services during different phases of the pandemic (pre lockdown, lockdown, post lockdown easing) have impacted on cancer care delivery, we sought weekly information for urgent cancer referrals for early diagnosis ('2-week wait' (2WW)), an indicator of both patient demand and health service supply, that is, how well the service is ensuring that individuals with suspicious symptoms are rapidly prioritised to the diagnostic cancer pathway and chemotherapy attendances (an indicator of supply and a proxy for possible adverse effects of the pandemic on the cancer treatment pathway). We employed the UK's Health Data Research Hub for Cancer (DATA-CAN)[21] to approach eight hospital trusts (in Leeds, London and Northern Ireland) and sought data from January 2019 to June 2020 to control for seasonal changes. Each hospital trust rapidly provided the requested data and permission to share these data in the public domain. We estimated the per cent change in weekly activity compared with the mean activity in 2019.

### Weekly near real-time death registration data
To estimate direct (among those infected) and indirect impact of the COVID-19 pandemic on deaths, we sought weekly counts of deaths in England and Wales from the Office for National Statistics (ONS), with causes classified by the ONS as COVID-19 deaths, non-COVID-19 deaths excluding cancer, and cancer deaths.

### Study population: primary care population-based cohort
To estimate pre-COVID-19 incidence and mortality in individuals with cancer, we used population-based electronic health records in England from primary care data from the Clinical Practice Research Datalink (CPRD) linked to the ONS death registration. We used this primary care data source because of the extensive information on comorbidities (which may be lacking in cancer registry data). The study population was 3 862 012 adults aged ≥30 years, registered with a general practice from 1 January 1997 to 1 January 2017, with at least 1 year of follow-up data. CPRD data are representative of the English population in terms of age, sex, mortality and ethnicity,[22–24] with extensive evidence of validity.[25] This study was performed as part of the CALIBER program (https://www.ucl.ac.uk/health-informatics/caliber). CALIBER is an open-access research resource consisting of information, tools and phenotyping algorithms available through the CALIBER portal (https://caliberresearch.org/portal).[26 27]

### Open-access definitions of disease using electronic health records
We defined non-fatal incident cases (as alive for at least 30 days following cancer diagnosis) and prevalent cases of cancer across 24 primary cancer sites according to previously validated CALIBER electronic health record phenotypes. Incident cancers were defined as new cancer diagnoses after the study entry into CPRD (baseline). Prevalent cancers were defined as cancer diagnoses recorded at any time prior to baseline. The cancers included: biliary tract, bladder, bone, brain, breast, cervix, colorectal, Hodgkin's lymphoma, kidney, leukaemia, liver, lung, melanoma,

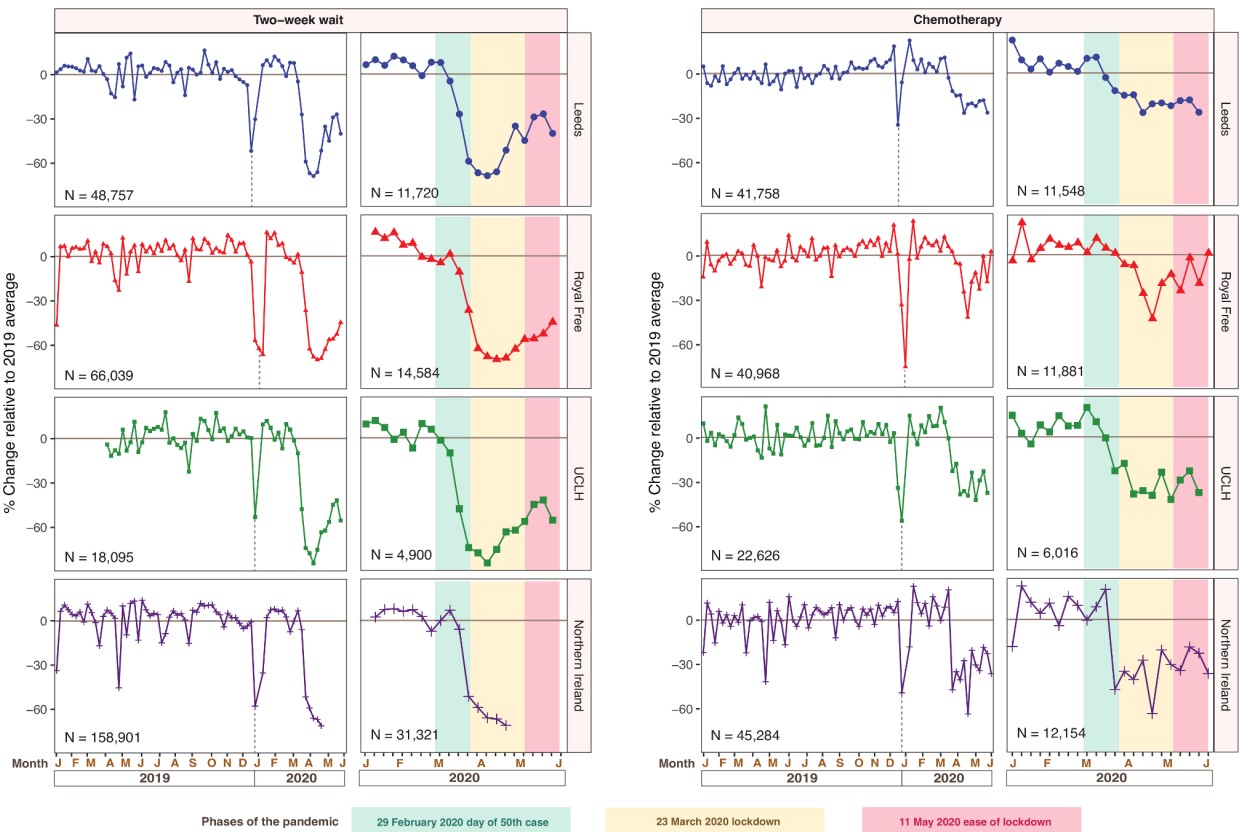

**Figure 1** Weekly hospital data (January 2019–June 2020) on changes in urgent referrals and chemotherapy clinic attendance from eight hospitals in the UK mapped to phases of the pandemic. Weekly changes from January 2020 to June 2020 were mapped to phases of the pandemic. Weekly values were plotted as percentage increase or decrease relative to the 2019 average. The data for Northern Ireland include five health and social care trusts (HSCs) that cover all health service provisions in Northern Ireland: Belfast HSC, Northern HSC, South Eastern HSC, Southern HSC and Western HSC. Vertical dotted lines indicate the Christmas bank holiday.

multiple myeloma, non-Hodgkin's lymphoma, oesophagus, oropharynx, ovary, pancreas, prostate, stomach, testis, thyroid and uterus.[28] Phenotype definitions of cancers and COVID-19-relevant comorbidities are available at https://caliberresearch.org/portal and have previously been validated.[29–32] Phenotypes were generated from hospital and primary care information recorded in primary care, using Read Clinical Terminology (V.2).

### Comorbidities relevant to COVID-19

We examined 15 comorbidity clusters, which include 40 non-malignant comorbidities defined by PHE as associated with severe and fatal COVID-19 infection.[17 18] We separately estimated the proportion of patients with each comorbidity at study entry (prevalent cancers), and at the date of the first diagnosis of incident cancer. The PHE list included chronic respiratory disease, chronic heart disease, immunocompromised individuals, HIV, use of corticosteroids, obesity, diabetes, chronic kidney disease, chronic liver disease, chronic neurological disorders and splenic disorders. A full list of the conditions we examined and their definitions are provided in online supplemental methods.

### Estimating incidence rates and 1-year mortality

We estimated incidence rates per 100 000 person-year and 1-year mortality in our study population. Estimated incidence rates for the number of new cancers by cancer site were compared with those for the UK from the International Agency for Research on Cancer and were found to be representative. We estimated baseline 1-year mortality risk following cancer diagnosis for both incident and prevalent cancers using Kaplan-Meier analyses stratified by cancer sites and number of (non-cancer) comorbid conditions (0, 1, 2 and 3+). We used the most recent 5 years of data (2012–2016) to estimate 1-year mortality.

### Estimating 1-year direct excess deaths

Excess deaths were estimated by applying relative risks (RRs) to the background 1-year mortality risk. Direct excess deaths (due to or with COVID-19) were modelled using the range of RRs (1.2, 1.5 and 2.0) previously reported in studies of cancer and COVID-19 deaths.[3 33] We applied these RRs to 10% of the population (the directly 'infected'), based on recent SARS CoV-2 seroprevalence estimates in the UK[34 35] and other countries.[36 37] Although the infection rate will change depending on the phase of the pandemic, we assumed an infection rate over 1 year in line with the first wave of the pandemic.

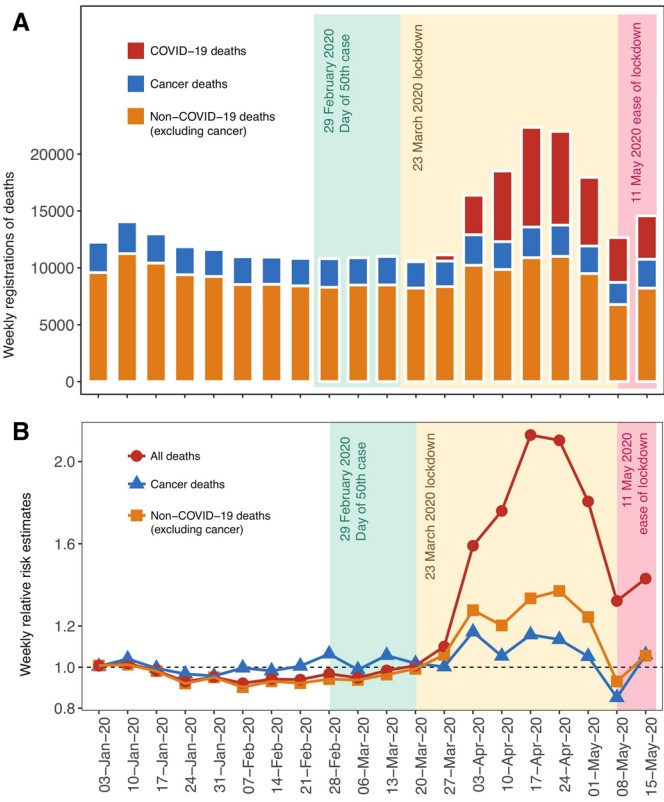

**Figure 2** Office for National Statistics data on weekly registrations of deaths in the England and Wales from 3 January 2020 to 15 May 2020. (A) Upper panel indicates the number of weekly deaths. (B) Lower panel indicates weekly changes in relative risk estimates calculated by comparing the current weekly deaths to 5-year weekly averages. Dates indicate week ending on a particular date.

### Estimating 1-year total (direct and indirect) excess deaths

Indirect excess deaths (due to pandemic-induced health service reconfiguration) were estimated by applying RRs for excess cancer deaths observed using ONS data, by taking the number of weekly cancer deaths from January 2020 divided by the weekly average over the last 5 years. We assumed that the effects of service change may not translate to an immediate increase in excess deaths. We have applied the RRs of 1.2, 1.5 and 2 to 40% (10% infected, 30% affected) and 80% (10% infected, 70% affected) of the population and modelled excess deaths over a 12-month period to capture medium-term effects. We chose this range of indirectly 'affected' population based on our real-time estimates of the degree of perturbation in cancer care during the pandemic and patient reports that clinical care had been cancelled during the pandemic for 53%–70% of patients with cancer or other conditions.[38]

To project the study estimates of excess deaths to the whole English population, we employed the 2018 population estimate, where the number of deaths is scaled up to a population of 35 407 313 individuals

aged 30 and above.[39] All analyses were performed using R (V.3.4.3).

## RESULTS
### Near real-time data on cancer care

Evaluating data from 291 792 people with suspected cancer and 150 636 patients with cancer attending for chemotherapy from January 2019 to June 2020, we initially characterised the pre pandemic basal level of activity (2019 average), including seasonal variations (figure 1). Using the date of the 50th patient diagnosed with COVID-19 as the starting point of the pandemic, we observed that urgent referrals fell by 70.4% (range: −68.7% to −84.3%), while chemotherapy attendances declined by 41.5% (range: −26.3% to −63.4%) (figure 1). To highlight these adverse impacts, we provided these data to chief medical officers in all four nations of the UK and the national director for cancer (England). We have also continued to provide regular updates of this intelligence to the Scientific Advisory Group for Emergencies. Since the NHS letter on 29 April 2020 restarting cancer and other services,[40] and since easing of lockdown (11 May 2020), there has been evidence of recovery for the urgent 2WW referrals (−55.4% to −40.0%; median=−44.5%), and chemotherapy attendances (−37.1% to 3.9%; median=−31.2%) (figure 1).

### Near real-time data on cancer, COVID-19 and other deaths

There were 1307 excess cancer deaths from 13 March to 15 May 2020 compared with the 5-year average based on weekly registration of deaths for England and Wales (figure 2A). We found an excess in cancer deaths with a peak in the week ending on 3 April 2020 with an RR of 1.17 (figure 2B). There were 41 105 COVID-19 deaths until 15 May 2020. For non-COVID-19 deaths (excluding cancers), we found that the peak occurred with an RR of 1.37 on 24 April 2020. The peak RR for all-cause deaths was 2.1, from the week ending 17 April 2020.

### Estimations on direct excess deaths by cancer site over 1 year

We estimated direct excess COVID-19 deaths based on a SARS CoV-2 infection rate of 10% and background 1-year mortality risks (figure 3A). For both incident and prevalent cancers combined, we estimated 1790, 4479 and 8957 direct excess deaths at RRs of 1.2, 1.5 and 2.0, respectively (figure 3B). Incidence rates for 24 cancer types were shown in online supplemental figure S1. Online supplemental figures S2 and S3 show the separate direct excess death estimates for incident and prevalent cancers.

### Estimations of total (direct and indirect) excess deaths by cancer site over a 1 year

When applying RRs of 1.2 or 1.5 to 40% (10% infected, 30% affected) of the population of people with cancer (both incident and prevalent cancers), we estimated 7165 and 17 910 total excess deaths, respectively (figure 3B). When applying these RRs to 80% (10% infected, 70%

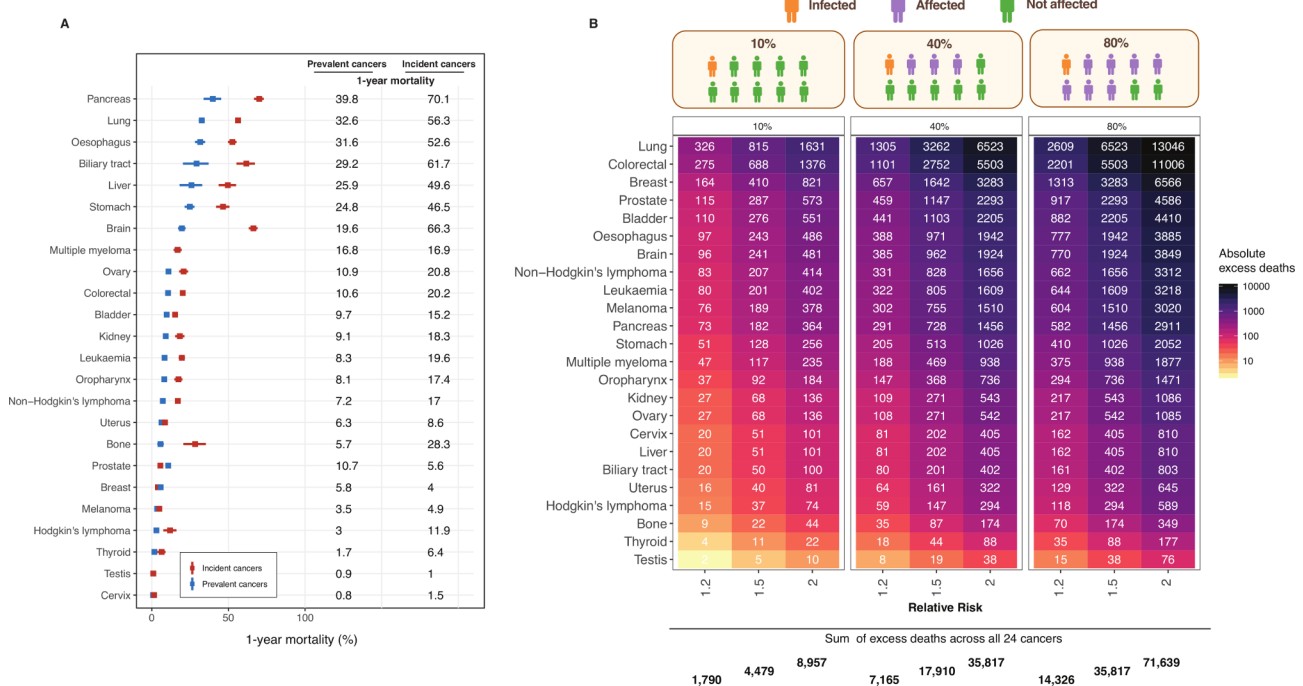

**Figure 3** Estimated total (direct and indirect) excess deaths by cancer site over a 1-year period. (A) 1-year mortality for incident and prevalent cancers. The whiskers are 95% CIs. (B) Total excess deaths were scaled up to the population of England aged 30+ consisting of 35 million individuals using England mortality estimates for both incident and prevalent cancers combined. We estimated direct excess deaths at a 10% infection rate. We estimated total (direct and indirect) excess deaths for 40% (10% infected, 30% affected) and 80% (10% infected, 70% affected) of the population.

affected) of the population of people with cancer, we estimated 14 326 and 35 817 total excess deaths, respectively (figure 3B). Online supplemental figures S2 and S3 show the separate total excess death estimates for incident and prevalent cancers.

### Comorbidities relevant to COVID-19 risk: prevalence and association with 1-year mortality

Comorbidities were common in people with incident cancer: hypertension (83313 (41.9%)), cardiovascular disease (55742 (28.0%)), chronic kidney disease (31935 (16.0%)), obesity (19589 (9.8%)), type 2 diabetes (18957 (9.5%)) and chronic obstructive pulmonary disease (COPD) (18373 (9.2%)) (online supplemental figure S4). Similar patterns were seen in prevalent cancers (online supplemental figure S5). Multimorbidity (≥1 comorbidity) was associated with a higher 1-year mortality (online supplemental figure S6) for incident cancers and (online supplemental figure S7) for prevalent cancers. For example, for incident colorectal cancer, 1-year mortality for 0, 1, 2 and 3+ comorbidities was 13.8%, 17.3%, 23.6% and 30.2%, respectively (online supplemental figure S6).

### Estimations of total (direct and indirect) excess deaths by cancer site and number of comorbidities over a 1-year period

To ascertain the influence of multimorbidity on total excess deaths, we provide estimates based on 40% (10% infected, 30% affected). For both incident and prevalent cancers, 78% of the predicted excess deaths occur in people with 1+ comorbidity. For example, at RR of 1.2,

there are 5622 excess deaths in those with 1+ comorbidity compared with 1567 in those with no comorbidities (total 7189) (figure 4). Even though the size of patient group in 0, 1, 2 and 3+ comorbidities declines (49.8%, 24.7%, 15.0% and 10.6%, respectively), the absolute numbers of excess deaths in each comorbidity group are similar, suggesting that patients with comorbidities contribute to a large proportion of excess deaths compared with those without non-cancer comorbidities. For example, at RR of 1.5, the numbers of total excess deaths for both incident and prevalent cancers were 3922, 4993, 4526 and 4542 in individuals with 0, 1, 2 and 3+ non-cancer comorbidities, respectively (figure 4). The findings for incident and prevalent cases are presented separately in online supplemental figures S8 and S9.

We share the underlying study estimates from this paper (online supplemental data) and provide an open-access tool for researchers to interact with the model (https://pasea.shinyapps.io/cancer_covid_app/).

### DISCUSSION
#### Statement of principal findings
To our knowledge, this is the first study with near real-time evidence of COVID-19's negative impact on cancer services at different phases of the pandemic, its potential to lead to significant excess deaths in people with cancer and the substantial role that comorbidities may play in these excess deaths.

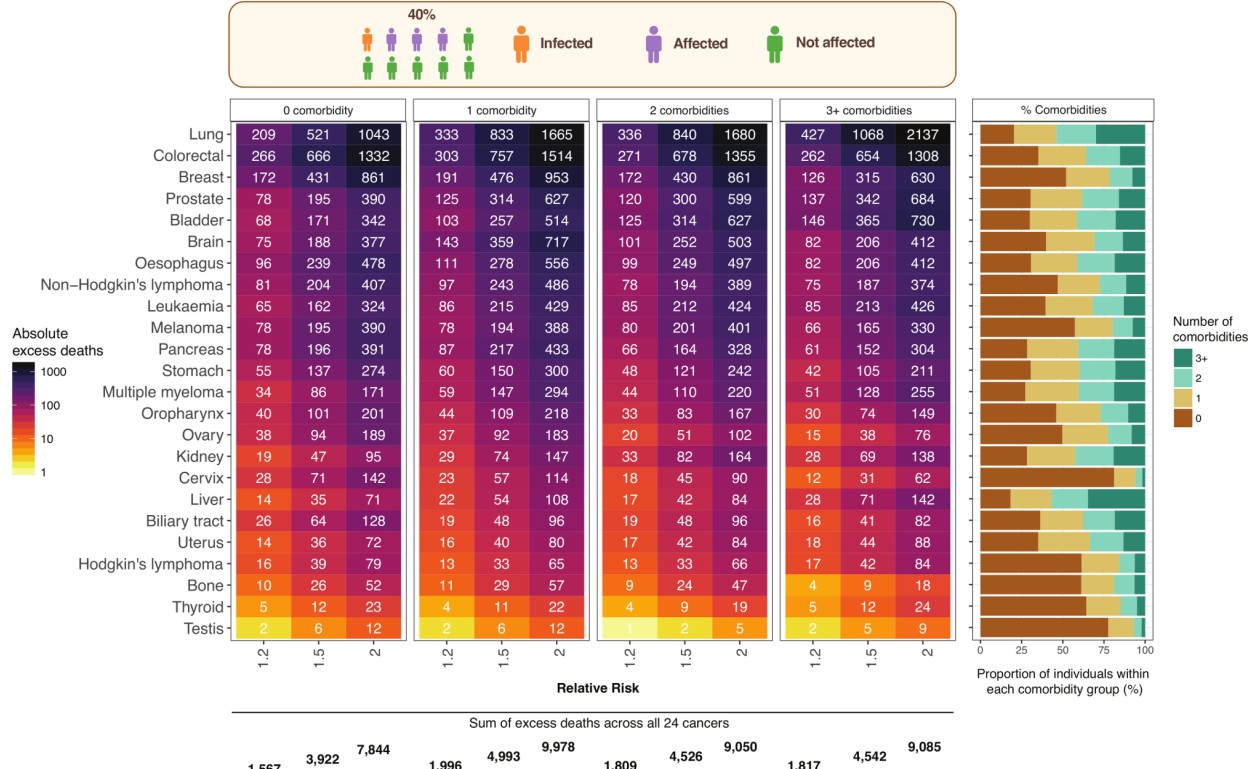

**Figure 4** Total (direct and indirect) excess deaths for both incident and prevalent cancers by cancer site and number of comorbidities over a 1-year period. Stacked bar chart indicates the proportion of individuals with 0, 1, 2 and 3+ comorbidities by cancer site. We estimated total excess deaths for 40% (10% infected, 30% affected) of the population. Total excess deaths were scaled up to the population of England aged 30+ consisting of 35 million individuals using England mortality estimates for both incident and prevalent cancers combined.

### Changes in cancer care at different phases of pandemic

We delineate both the nadir and the incomplete recovery of the UK cancer services that have resulted from the COVID-19 pandemic. We observed profound declines in urgent 2WW referrals for early cancer diagnosis, which have not returned to pre-COVID-19 levels. These may reflect patients' deciding not to seek care due to the perceived risk of infection, but may also be in part due to difficulty in securing appointments due to reprioritised health systems.[19] An unintended consequence of this reprioritisation may be excess deaths due to delayed diagnoses, increased emergency presentations, more advanced stage at presentation and changes in care pathways that adversely affect outcomes. We also observed large declines in chemotherapy attendance, presumably reflecting capacity/resources being redirected to care for infected patients (eg, to intensive care) and the desire of clinicians and patients to minimise the risks of COVID-19 for susceptible patients with cancer.[10]

### Direct (COVID-19) excess deaths

It is important to note that our model estimates deaths additional to those that would be expected (without COVID-19) in people with cancer. At an RR of 2, we estimate about 9000 direct COVID-19 excess deaths in 1 year in people with cancer but acknowledge there is uncertainty in this estimate. There is increasing concern that

those discharged from hospital with COVID-19 may have long-term (including fatal) sequelae.

### Total (direct and indirect) excess deaths

Based on our observations regarding the adverse effects of cancer service reprioritisation, we consider a proportion affected by the pandemic of 40% plausible, if perhaps somewhat conservative. But, given that adverse effects could be more profound (our 2WW referrals data, for example, would suggest this), we present excess deaths for a range of both 40% and 80%. Adding credibility to our estimates, in a survey in April 2020 of 17 000 UK adults, 56% of patients with cancer reported that the NHS had cancelled their treatment.[38] Overall, we conservatively estimate, at RR of 1.5, that 17 910 total excess deaths for 1 year will occur in patients with cancer, but this could rise to 35 817. We note the degree of uncertainty in the observed RR at different points in the pandemic. Patients affected by changes in cancer services in March–June 2020 may not necessarily directly contribute to an increase in excess indirect deaths in these 4 months, as the effects on health and mortality outcomes are more likely to occur in a longer time frame.

### Importance of multimorbidity

We demonstrate that the majority (78%) of excess deaths in people with cancer during the COVID-19 pandemic occur

in people with at least one comorbidity. While many of these comorbidities are treatable, services for these conditions have also been affected by the pandemic. For example, 65% of patients with hypertension and 70% of patients with diabetes reported that the NHS had recently cancelled their care, as captured in the same April 2020 survey noted above.[38] Importantly, the pandemic prompts new questions about which patients with cancer are most vulnerable and how best to mitigate an individual's personal risk.

### Strengths of this study

There are three major strengths of this study. First, the acquisition and deployment of near real-time data to signal the significant adverse impact of the COVID-19 pandemic on cancer services and how this has profound implications for cancer diagnostic and treatment pathways. These data were also used to inform and enhance our existing model that estimates excess mortality due to the pandemic. Second, we provide a pan-cancer comorbidity atlas using a population-based 3.8 million primary care cohort to underpin estimates of the additional adverse effect of multimorbidity in patients with cancer; cancer registry data tend to lack this more comprehensive information. Third, we provide separate estimates of excess deaths for prevalent cancers and incident (newly diagnosed) cancers, because these represent different patterns of risk, treatment priorities and roles of general practitioner and oncologist.

### Weaknesses of this study

Our model has important limitations. First, there is a lack in the literature of studies on clinical cohorts of patients with cancer investigating all-cause mortality rates in those with and without infection; such studies are needed in order to obtain better estimates of the direct effects of the pandemic. Second, the primary care health records we used may have missed some cases of cancer and thus underestimated incidence.[41] If so, our estimates of excess deaths may be conservative. The NHS has national linked hospital admissions and cancer registration data with information on stage and details of surgical, chemotherapeutic and radiotherapy treatment of cancer. However, information governance for such data can take months to secure, making data-enabled research and time-sensitive responsive service improvement difficult. Third, we did not have access to data on children. Fourth, we only have access to empirical cancer service change data from eight hospitals in the UK. While the data may be a representative sample of the UK population, and patterns of decline in service change is corroborated in another study,[42] more widespread access to other trusts may be beneficial to ascertain national and regional effects.

### Implications for clinicians and policymakers

Our study may inform decision-making at three levels. First, from a healthcare policy and healthcare implementation perspective, it is clear that the NHS cannot simply be 'switched on' again at full capacity for hospital or primary care services as there will be a significant backlog of untreated patients, with waiting lists predicted to expand to 10 million patients. Data published on 13 June 2020 indicate ~100 000 'missing' cancer referrals in April 2020 alone.[7] More granular weekly intelligence from the centres contributing data to this study suggests that this negative impact will continue for at least 6–9 months, placing many more patients at risk.

Second, there are currently no accessible national systems available for near real-time data on care and outcomes of patients with cancer. Our study suggests that we should expand our near real-time data approach across the UK to collect actionable information on (1) death certification—in particular distinguishing the contribution of cancer, comorbid conditions and COVID-19 to death; (2) cancer health services activity data, to monitor how changes at each phase of the pandemic (including clearing backlogs for under referral, under diagnosis and under treatment) might influence future health outcomes and (3) treatment services data for non-malignant comorbidities of patients with cancer, such as cardiovascular disease, diabetes and hypertension.

Third, with knowledge of mortality risk based on type of cancer, age and comorbidities that we provide in an online format (https://pasea.shinyapps.io/cancer_covid_app/), supplemented with local knowledge of health service resilience, we propose that weekly indicators and warnings for vulnerable patients with cancer with multimorbidity could be provided. Using this intelligence, treatment prioritisation as we resume cancer services could be enhanced by patient-specific risk/benefit assessments, which include multimorbidity, particularly in situations where treatment provision outweighs non-treatment/safety issues related to COVID-19.[19]

### Unanswered questions and future research

There are important areas for further research. First, there is a need for long-term (1–5 years) monitoring of the extent to which patients with cancer experience excess mortality due to the pandemic. We chose a 1-year time horizon, because the adverse consequences on health are likely to extend beyond the initial wave of the pandemic. But its impact on excess mortality in patients with cancer, particularly those whose diagnosis/treatment is delayed, may take years to understand. The specific impact of paused cancer screening, particularly for breast and colorectal cancer, may be profound. The social and psychological consequences of physical distancing on mortality may also be particularly important in cancer,[43 44] while international studies across 75 countries signpost how unemployment negatively impacts mortality in patients with cancer.[45] Hence, the socioeconomic effects of the current pandemic are likely to last for a considerable period beyond 1 year.[46] As new empirical data become available on heath service, social/psychological and economic changes, our model can better specify the proportion and type of patients with cancer thus affected and look to develop appropriate mitigation strategies.

## CONCLUSION

We mobilised usually inaccessible near real-time hospital data to quantify the immediate adverse impacts of the COVID-19 pandemic on cancer services, on people who may demonstrate symptoms of cancer and on patients who are being treated for cancer. The marked reductions observed in the demand for, and supply of, cancer services have only partially recovered with lockdown easing. Such perturbations in cancer care may contribute, over a 1-year time horizon, to substantial excess mortality among people with cancer and multimorbidity. There is an urgent need to better understand and mitigate these excess mortality risks, some of which may be revealed only over the longer term.

**Author affiliations**
[1]Institute of Health Informatics, University College London, London, UK
[2]Health Data Research UK, University College London, London, UK
[3]Barts Health NHS Trust, The Royal London Hospital, Whitechapel Rd, London, UK
[4]DATA-CAN, Health Data Research UK hub for cancer hosted by UCLPartners, London, UK
[5]Leeds Institute of Medical Research, University of Leeds, Leeds, UK
[6]Leeds Teaching Hospitals NHS Trust, Leeds, UK
[7]University College London Hospitals NIHR Biomedical Research Centre, London, UK
[8]The Alan Turing Institute, London, UK
[9]Institute of Cardiovascular Science, University College London, London, UK
[10]University College London Hospitals NHS Trust, London, UK
[11]Division of Infection and Immunity, University College London, London, UK
[12]Department of Hematology, University College London Cancer Institute, London, UK
[13]University College London Cancer Institute, London, UK
[14]Royal Free NHS Foundation Trust, London, UK
[15]Division of Genetics and Epidemiology, Institute of Cancer Research, London, UK
[16]Northern Ireland Cancer Network, Northern Ireland, UK
[17]Barts Liver Centre, Blizard Institute, Queen Mary University of London, London, UK
[18]Office for National Statistics, London, UK
[19]Leeds Institute of Health Sciences, University of Leeds, Leeds, UK
[20]UCLPartners Academic Health Science Partnership, London, UK
[21]Centre for Cancer Outcomes, University College London Hospitals NHS Foundation Trust, London, UK
[22]UCL Great Ormond Street Institute of Child Health, University College London, London, UK
[23]Conflict and Health Research Group, Institute of Cancer Policy, King's College London, London, UK
[24]Patrick G Johnston Centre for Cancer Research, Queen's University Belfast, Belfast, UK

**Acknowledgements** We thank Tony Hagger, Shiva Thapa, Mohammed Emran, Cara Anderson, Louise Herron, Joy Beaumont, Maurice Loughrey, Philip Melling and Lee Cogger for their help on collating data on urgent cancer referrals and chemotherapy attendances. We thank the Health Data Research UK DATA-CAN Patient and Public Involvement and Engagement panel for critical feedback on the manuscript. We thank Charles Swanton for his valuable comments on the manuscript. This study is based in part on data from the Clinical Practice Research Datalink obtained under licence from the UK Medicines and Healthcare products Regulatory Agency. The data are provided by patients and collected by the NHS as part of their care and support. Mortality data are from the Office for National Statistics.

**Contributors** Research question: AL and HH. Funding: AL, AB, ML and HH. Study design and analysis plan: AL, LP, AB, MK, WHC and HH. Preparation of data, including electronic health record phenotyping in the CALIBER portal: AL, LP and SD. Provision of weekly hospital data: GH, KP-J, MF, DH, ML, KB and CD. Statistical analysis: AL, LP, WHC and MK. Drafting initial versions of manuscript: AL, ML and HH. Drafting final versions of manuscript: AL, GH, CD, ML and HH. Critical review of early and final versions of manuscript: AL, LP, AB, GH, SD, WHC, MK, BW, DP, MN, DL, DH, MF, CT, NF, KB, GF, TE, VN, BH, RDN, MC, MJ, KP-J, RS, CD, ML and HH. Joint second authors: LP, AB and GH.

**Funding** DATA-CAN (MC_PC_19006) is part of the Digital Innovation Hub Programme, delivered by HDR UK and funded by UK Research and Innovation through the government's Industrial Strategy Challenge Fund. AL is supported by funding from the Wellcome Trust (204841/Z/16/Z), National Institute for Health Research (NIHR) University College London Hospitals Biomedical Research Centre (BRC714/HI/RW/101440), NIHR Great Ormond Street Hospital Biomedical Research Centre (19RX02) and the Health Data Research UK Better Care Catalyst Award. AB is supported by research funding from NIHR, British Medical Association, Astra-Zeneca and UK Research and Innovation. KP-J is supported by the NIHR Great Ormond Street Hospital Biomedical Research Centre. CD and KP-J are funded by UCLPartners. HH is an NIHR senior investigator and is funded by the NIHR University College London Hospitals Biomedical Research Centre, supported by Health Data Research UK (grant No. LOND1), which is funded by the UK Medical Research Council, Engineering and Physical Sciences Research Council, Economic and Social Research Council, Department of Health and Social Care (England), Chief Scientist Office of the Scottish Government Health and Social Care Directorates, Health and Social Care Research and Development Division (Welsh Government), Public Health Agency (Northern Ireland), British Heart Foundation, Wellcome Trust, the BigData@ Heart Consortium, funded by the Innovative Medicines Initiative-2 Joint Undertaking under grant agreement No. 116 074.

**Competing interests** ML has received honoraria from Pfizer, EMD Serono and Roche for presentations unrelated to this research, and an unrestricted educational grant from Pfizer for research unrelated to the research presented in this paper. AB has received research funding from AstraZeneca. MF has received research funding from AstraZeneca, Boehringer Ingelheim, Merck and MSD and honoraria from Achilles, AstraZeneca, Bayer, Boehringer Ingelheim, Bristol-Meyers Squibb, Celgene, Guardant Health, Merck, MSD, Nanobiotix, Novartis, Pharmamar, Roche and Takeda for advisory roles or presentations unrelated to this research. GF receives funding from companies that manufacture drugs for hepatitis C virus (AbbVie, Gilead, MSD) and is a consultant for GSK, Arbutus and Shionogi in areas unrelated to this research.

**Patient consent for publication** Not required.

**Ethics approval** The study was approved by the MHRA (UK) Independent Scientific Advisory Committee (20_074R2), under Section 251 (NHS Social Care Act 2006).

**Provenance and peer review** Not commissioned; externally peer reviewed.

**Data availability statement** Data may be obtained from a third party and are not publicly available. Data used in this study were accessed through the Clinical Practice Research Datalink that is subject to protocol approval by an Independent Scientific Advisory Committee and cannot directly be shared. All results are reported in the manuscript and no additional data are available.

**ORCID iDs**
Alvina G Lai http://orcid.org/0000-0001-8960-8095
Amitava Banerjee http://orcid.org/0000-0001-8741-3411
Bryan Williams http://orcid.org/0000-0002-8094-1841
Clare Turnbull http://orcid.org/0000-0002-1734-5772
Vahe Nafilyan http://orcid.org/0000-0003-0160-217X
Harry Hemingway http://orcid.org/0000-0003-2279-0624

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
