## [Reviewer comments · BMJ Open]

ARTICLE DETAILS

TITLE (PROVISIONAL)	Estimated impact of the Covid-19 pandemic on cancer services and excess 1-year mortality in people with cancer and multimorbidity: near-real-time data on cancer care, cancer deaths and a population-based cohort study
AUTHORS	Lai, Alvina; Pasea, Laura; Banerjee, Amitava; Hall, Geoff; Denaxas, S; Chang, Wai Hoong; Katsoulis, M; Williams, Bryan; Pillay, Deenan; Noursadeghi, Mahdad; Linch, David; Hughes, Derralynn; Forster, Martin; Turnbull, Clare; Fitzpatrick, Natalie; Boyd, Kathryn; Foster, Graham; Enver, Tariq; Nafilyan, Vahe; Humberstone, Ben; Neal, Richard; Cooper, Matt; Jones, Monica; Pritchard-Jones, Kathy; Sullivan, Richard; Davie, Charlie; Lawler, Mark; Hemingway, Harry

VERSION 1 – REVIEW

REVIEWER	Zhi Ven Fong, MD, MPH Massachusetts General Hospital, United States.
REVIEW RETURNED	06-Oct-2020

GENERAL COMMENTS	The authors should be commended for the amount of effort placed in revising the manuscript. The paper in its current draft is more complete, and makes for a clearer illustration of the valuable findings that the authors have noted for the journal's readers. It's also a clearer read.
--

VERSION 1 – AUTHOR RESPONSE

This paper has been accepted for publication.

As requested, figures and supplementary figures citations are now all in ascending orders.